# Accelerating Visual Sparse-Reward Learning with Latent Nearest-Demonstration-Guided Explorations

**Ruihan Zhao**[1] **Ufuk Topcu**[1†] **Sandeep Chinchali**[1†] **Mariano Phielipp**[2†]

[1]The University of Texas at Austin  [2]Intel AI Lab

**Abstract:** Recent progress in deep reinforcement learning (RL) and computer vision enables artificial agents to solve complex tasks, including locomotion, manipulation, and video games from high-dimensional pixel observations. However, RL usually relies on domain-specific reward functions for sufficient learning signals, requiring expert knowledge. While vision-based agents could learn skills from only sparse rewards, exploration challenges arise. We present **La**tent **N**earest-demonstration-guided **E**xploration (LaNE), a novel and efficient method to solve sparse-reward robot manipulation tasks from image observations and a few demonstrations. First, LaNE builds on the pre-trained DINOv2 feature extractor to learn an embedding space for forward prediction. Next, it rewards the agent for exploring near the demos, quantified by quadratic control costs in the embedding space. Finally, LaNE optimizes the policy for the augmented rewards with RL. Experiments demonstrate that our method achieves state-of-the-art sample efficiency in Robosuite simulation and enables *under-an-hour* RL training from scratch on a Franka Panda robot, using only a few demonstrations. *

**Keywords:** Computer Vision, Sparse Reward, RL from Demonstrations

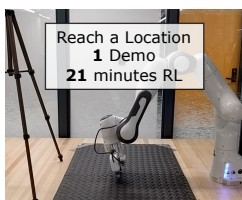 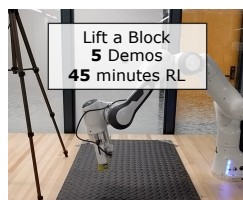 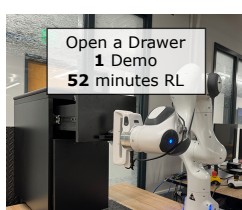 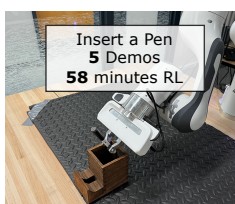

Figure 1: LaNE enables under-an-hour RL training from scratch on a Franka Panda arm from image observations and sparse rewards, utilizing only a few demonstrations. LaNE achieves unparalleled sample efficiency by learning an embedding space to quantify state proximity and reward explorations close to the demonstrations.

## 1 Introduction

Deep reinforcement learning (RL) is a versatile approach that learns from interaction data, often without an explicit, hand-coded dynamics model. Through environmental interactions, RL agents can learn optimal policies from dense or sparse reward feedback. State-of-the-art approaches allow learning policies for discrete actions and continuous action spaces while taking either low-dimensional state vectors or high-dimensional sensor readings [1, 2, 3, 4].

However, applying deep RL to real-life domains, including real-hardware robot learning, remains difficult. One challenge is the need to reliably track the complete system state [5]. A policy that directly maps images to optimal actions could alleviate such engineering challenges: data augmentation and self-supervised learning have enabled policy learning from image observations with high

---

[†]Equal advising.
[*]Project page: https://philipzrh.com/lane/.

8th Conference on Robot Learning (CoRL 2024), Munich, Germany.

sample efficiency [6, 7, 8, 9]. Meanwhile, it is also hard to assign informative rewards in a scalable way. Reward engineering requires domain-specific knowledge: popular simulated environments provide optional dense reward functions based on heuristics [10, 11] but rely on state readings not readily available in the real physical system. Thus, there has been a lot of effort to help RL agents explore more effectively in environments with sparse or no rewards [12, 13, 14].

We present **La**tent **N**earest-demonstration-guided **E**xploration (LaNE), a novel approach to tackle the exploration challenge in image-based control tasks with sparse rewards. LaNE builds on the idea of learning from demonstrations (LfD) and draws inspiration from reward shaping [15, 16, 17, 18, 19, 20, 21, 22, 23, 24, 25]. Our main insight is that each step in the demonstration can be considered a subgoal, and the agent should be credited for reaching a similar state (see Fig. 2). Our approach presents two main contributions:

1. We define a distance measure among image observations by learning a lower-dimensional latent space. Specifically, we train a Variational Autoencoder (VAE) so that the resulting latent space's forward dynamics are locally linear. A quadratic control cost in this space effectively identifies nearby states, whereas pre-trained embeddings from even a state-of-the-art computer vision model are insufficient.

2. We propose a systematic way to provide dense reward signals in sparse-reward tasks under the LfD paradigm. When exploring near the demonstrations, the RL agent receives additional *task-progress-informed* rewards. The augmented reward function also derives bounded value functions, significantly improving training stability.

LaNE is independent of the underlying RL algorithm, and we use Soft Actor-Critic [1] in this paper. In Robosuite simulation [11], our method significantly improves sample efficiency when learning long-horizon, sparse-reward visual manipulation tasks. On real hardware, LaNE enables learning various manipulation tasks from scratch with a Franka Panda arm, each with under an hour of training and only one to five demonstrations (see Fig. 1).

## 2 Related Work

Exploration is a known challenge in deep RL, especially in sparse-reward environments [26]. One solution is to guide exploration with domain knowledge. In reward shaping, intermediate rewards can be added at important checkpoint locations [11], derived from physics knowledge [27], or learned from human annotations [28]. Aside from external guidance, other works aim to improve the agents' intrinsic exploration behavior. Maximum entropy RL balances exploration and exploitation by encouraging high policy entropy [1]. Hierarchical RL and intrinsically motivated skill learning are also potential remedies [12, 29]. Our work aims to alleviate the need for domain expertise in the reward-shaping approach by using a few demonstrations.

Learning from demonstrations (LfD) has proven helpful in expediting RL, especially in sparse reward settings. Prior works have introduced various auxiliary training objectives. Behavior cloning [21, 23] and supervised $Q$-value updates [18] can effectively learn from optimal demonstrations. [30] employ an information theoretic approach to guide policy distributions. GAIL [31], AIRL [32] and DAC [33] perform adversarial training to distinguish expert and policy rollouts. [17] and [20] learn potential functions from value estimates, allowing imperfect or mismatched demonstrations to be used. CoDER [34] performs contrastive learning to pre-train the image encoders. Other methods perform warm start with scripted or behavior-cloned policies [14, 22]. Our method adopts the LfD paradigm for efficient learning and is compatible with sub-optimal demonstrations.

Aside from more efficient exploration and LfD, model-based RL methods aim to improve efficiency by generating new synthetic experiences with learned world models. DreamerV2 [6] learns an accurate discrete world model from high-dimensional image inputs, enabling human-level RL performance in Atari games. Modem [35] combines model-based learning with demonstrations by over-sampling demonstrated data to form a behavior prior. Our method LaNE also learns a forward

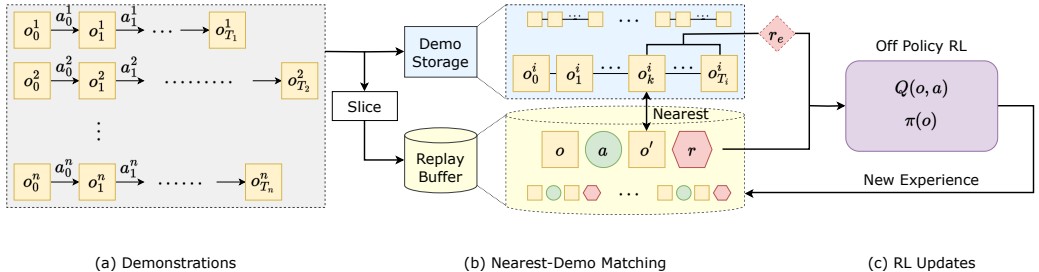



(a) Demonstrations       (b) Nearest-Demo Matching       (c) RL Updates



Figure 2: **La**tent **N**earest-demonstration-guided **E**xploration (LaNE) augments the sparse task reward with a dense exploration reward in vision-based RL. (**Left**) We utilize variable-length demonstrations, each consisting of observations $o_t^i$ and actions $a_t^i$. (**Middle**) A dense exploration reward $r_e$ is given when a transition lands sufficiently close to a demonstration and is discounted based on its distance to the goal. (**Right**) Using the combined reward signal, the RL agent learns to map a sensor observation $o$ to an action $a$.

dynamics model but only as an auxiliary task for representation learning. Hence, it can be integrated with both model-based and model-free RL methods.

Finally, a rich collection of prior work has studied ways to learn better representations from high-dimensional image observations. RAD and DrQ [8, 9, 36] perform data augmentation on image observations to promote task-relevant features. CURL and CoDER [7, 34] use contrastive learning as a self-supervised auxiliary objective for feature learning. TCC and TCN [37, 38] perform self-supervised learning from video sequences. Recently, foundation models have also proven beneficial for learning generalizable agents [39]. Our method builds on a state-of-the-art foundation model, DINOv2 [40], and uses data augmentation to learn a robust embedding space.

## 3 Problem Setting

We tackle the challenges of vision-based RL in long-horizon, sparse reward tasks, where the agent only receives a positive reward $r_{done}$ at task completion while getting a constant negative reward $r_{live}$ everywhere else. The interpretation of such a reward function is that the agent gets a high reward only at task completion but is penalized for the trajectory length. Formally, letting $\mathcal{G}$ denote the set of goal states, we define the reward function as follows:

$$r(s, a, s') = \begin{cases} r_{done} > 0 & s' \in \mathcal{G} \\ r_{live} < 0 & \text{otherwise.} \end{cases} \tag{1}$$

The sparse reward function reduces the need for expert knowledge or human intervention, making it much easier to implement in a real-world environment, but makes exploration hard in training.

## 4 Method

We present **La**tent **N**earest-demonstration-guided **E**xploration, an efficient RL algorithm centered around a few demonstrations to tackle the exploration challenge with sparse-reward learning. The core idea is to provide additional task-progress-aware dense rewards when the agent is close to the demonstrations. We learn a structured embedding space to quantify state proximity from image observations by learning a latent dynamics model as an auxiliary task. The cleverly designed augmented reward function also derives bounded value functions, enabling us to perform value clipping and greatly enhance training stability.

### 4.1 Reinforcement Learning from Demonstrations

LaNE utilizes a demonstration set $\mathcal{D}$ consisting of $n$ successful trajectories of observations and actions. Each demonstration trajectory $i$ may have a different length $T_i$, but must terminate in the goal set $\mathcal{G}$. We assume the demonstrations to come from a human operator or a heuristic controller

and, thus, can be sub-optimal. We formalize the notations as follows: $\mathcal{T}^i$ denotes the trajectory for demonstration $i$. $s_t$ is the underlying true environment state, and $o_t$ is the high-dimensional observation, such as images. $a_t$ is the action taken by the demonstrator. Note that the RL algorithm cannot access the ground truth state $s_t$ in the demonstrations, but only observations, as shown below:

$$\begin{aligned}
\mathcal{D} &= \{\mathcal{T}^1, \mathcal{T}^2 \cdots \mathcal{T}^n\} \\
\mathcal{T}^i &= (o_0^i, a_0^i, o_1^i, a_1^i, \cdots o_{T_i-1}^i, a_{T_i-1}^i, o_{T_i}^i) \\
&\forall i, s_{T_i}^i \in \mathcal{G}.
\end{aligned} \tag{2}$$

The demonstrations are stored in two forms. First, the trajectory form records the steps to success from each state, allowing us to discount the exploration reward according to task progress, as detailed in Section 4.3. Next, they are sliced into experience tuples $(o, a, o', r, d)$ and placed in a replay buffer $\mathcal{B}$ for representation learning and off-policy RL updates. Here, $o'$ denotes the next observation after $o$. $r$ is the sparse reward as defined in Equation 1, and $d$ is a Boolean variable indicating episode termination. LaNE uses a first-in-first-out replay buffer $\mathcal{B}$ with capacity for a limited number of transitions, but the demonstrations are always retained to ensure sufficient reward signal.

## 4.2 Augmentation-Invariant Distance Measure

The key technical challenge behind LaNE is to find a demonstration state closest to the agent's current state and quantify its proximity. Computing the distance between two states from their respective image observations is nontrivial: two drastically different states might only differ by a few pixels. Conversely, the same underlying state might appear very different in two images due to task-irrelevant background features. To solve this, we embed the images into a low-dimensional latent space to obtain a viable distance measure. Inspired by Embed to Control (E2C) [41], we train a VAE [42] and enforce a locally linear dynamics model to regularize the structure of the latent space. The locally-linear dynamics model captures our goal for the latent space to be temporally consistent since we have a multi-step control task.

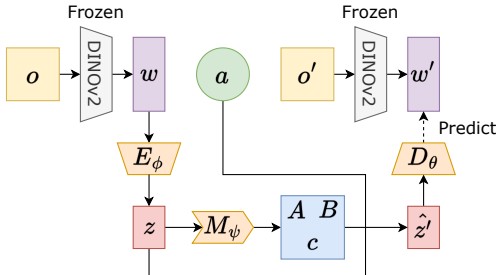

Figure 3: LaNE learns a latent space with locally linear dynamics. Given a transition tuple $(o, a, o')$, the observations are first encoded by a frozen DINOv2 model into $w$ and $w'$. Next, the encoder $E_\phi$ further embeds $w$ into a low-dimensional latent state $z$. The forward model $M_\psi$ predicts the transition matrices $A$, $B$ and offset $c$. Finally, the decoder $D_\theta$ reconstructs $w'$ from the predicted $\hat{z}'$, where $\hat{z}' = Az + Ba + c$. The trainable modules $E_\phi$, $M_\psi$, and $D_\theta$ are colored in orange.

Our method differs from E2C in three key ways: **1.** LaNE leverages the feature extractor from a strong pre-trained image model by embedding and predicting the frozen DINOv2 [40] features. As we find with ablation studies in Section 5.2, learning on top of DINOv2 features $w$ is superior to learning directly from pixels $o$. **2.** We learn a latent space robust to pixel-space perturbations. It has been shown that data augmentation is crucial for efficient and robust image-based reinforcement learning [8, 9, 34, 36]. Hence, we apply a random data-augmentation function $f(\cdot)$ during representation learning. **3.** Our RL policy network uses a separate CNN encoder to learn in an unconstrained embedding space and benefit from low inference latency. Overall, the trainable components include the encoder $E_\phi$, the decoder $D_\theta$, and the transition model $M_\psi$.

LaNE optimizes a variational lower bound (ELBO) objective across transition tuples $(o, a, o')_i$ sampled from the replay buffer. We assume the latent states $z$ form a unit Gaussian prior $p(\mathbf{z}) = \mathcal{N}(0, \mathcal{I})$. The encoded distributions $q(\mathbf{z} \mid w)$ and decoded distribution $p(\mathbf{w} \mid z)$ are also modeled with Gaussian distributions. During training, the data-augmented observation features are encoded into their latent distributions whose mean $\mu$ and diagonal covariance matrix $\Sigma$ are predicted by the encoder network $E_\phi$:

$$\begin{aligned}
z &\sim q_\phi(\mathbf{z} \mid w) = \mathcal{N}(\mu, \Sigma), \text{ where } (\mu, \Sigma) = E_\phi(w), \ w = \texttt{DINOv2}(f(o)) \\
z' &\sim q_\phi(\mathbf{z}' \mid w') = \mathcal{N}(\mu', \Sigma'), \text{ where } (\mu', \Sigma') = E_\phi(w'), \ w' = \texttt{DINOv2}(f(o')).
\end{aligned} \tag{3}$$

The one-step forward model in the latent space is locally linear in the state and action, whose parameters (matrices $A$, $B$ and offset $c$) depend on the starting state, as predicted by the latent transition model $M_\psi$. Prior work shows that a latent linear dynamics model is tractable to learn but provides modeling flexibility through local linearity [41]. The linear transition model allows the prediction of the next step latent distribution using the current distribution and action as follows:

$$\hat{z}' = Az + Ba + c, \text{ where } (A, B, c) = M_\psi(z) \tag{4}$$

$$q_\psi(\hat{\mathbf{z}}' \mid z, a) = \mathcal{N}(\hat{\mu}', \hat{\Sigma}'), \text{ where } \hat{\mu}' = A\mu + Ba + c, \ \hat{\Sigma}' = A\Sigma A^T. \tag{5}$$

Finally, the decoder $D_\theta$ reconstructs the next step observation embedding from the predicted next step latent vector:

$$\hat{w}' = D_\theta(\hat{z}'). \tag{6}$$

The encoder $E_\phi$, decoder $D_\theta$, and transition model $M_\psi$ are updated jointly using a combined loss with three terms. First, we want the sampled starting latent state $z$ to be reconstructed back to the original image features $w$. Similarly, as we pass the sample through the dynamics model, the resulting latent state prediction $\hat{z}'$ should be reconstructed back to $w'$. Finally, to ensure the latent dynamics model is consistent over multiple steps, we want the predicted distribution $q_\psi(\hat{\mathbf{z}}'|w, a)$ and encoded distribution $q_\phi(\mathbf{z}'|w')$ to be similar. Formally, we write the overall training objective $\mathcal{L}$ as follows, where $\lambda$ is a hyper-parameter for weighing the two loss terms:

$$\mathcal{L}_{\text{ELBO}} = \mathop{\mathbb{E}}_{z \sim q_\phi, \hat{z}' \sim q_\psi} \left[ -\log p(w|z) - \log p(w'|\hat{z}') \right] + D_{\text{KL}}\left( q_\phi(\mathbf{z} \mid w) \,\Big\|\, p(\mathbf{z}) \right) \tag{7}$$

$$\mathcal{L}_{\text{dynamics}} = \mathop{\mathbb{E}}_{z \sim q_\phi} \left[ D_{\text{KL}}\left( q_\psi(\hat{\mathbf{z}}' \mid z, a) \,\Big\|\, q_\phi(\mathbf{z}' \mid w') \right) \right] \tag{8}$$

$$\mathcal{L} = \mathop{\mathbb{E}}_{(o,a,o') \in \mathcal{B}} \left[ \mathcal{L}_{\text{ELBO}} + \lambda \mathcal{L}_{\text{dynamics}} \right]. \tag{9}$$

In essence, we minimize the reconstruction error for the VAE using the ELBO objective (term 1) and the forward prediction error in the latent space using KL divergence (term 2).

The learned latent space allows us to define a dynamics-aware distance measure between observations. Specifically, for two observations $o_1$ and $o_2$, we define the **A**ugmentation-invariant **D**istance **M**easure (ADM) to be the root quadratic cost between the augmented and *encoded* states $z_1$ and $z_2$:

$$d(o_1, o_2) := ((z_1 - z_2)^T Q (z_1 - z_2))^{\frac{1}{2}}. \tag{10}$$

Our design echoes the quadratic cost function commonly used in optimal control. Using an identity weighting matrix $Q = I$ further simplifies ADM to the Euclidean distance in the latent space. We apply this simplification in our experiments, but other weighting matrices could be useful when the agent observes both images and proprioceptive states.

## 4.3 Demonstration-Guided Exploration

We propose a systematic reward-engineering approach to credit the agent for staying close to demonstrations. Given an experience tuple $(o, a, o', r, d)$, we assign an additional exploration reward $r_e^*$ if $o'$ is sufficiently close to a demonstrated state, up to a distance threshold $\epsilon$, which is dynamically computed. We define $\epsilon$ as the average distance between consecutive demonstration observations:

$$\epsilon := \mathop{\mathbb{E}}_{i,t} \left[ d(o_t^i, o_{t+1}^i) \right], \qquad o_t^i, o_{t+1}^i \in \mathcal{D}. \tag{11}$$

The threshold $\epsilon$ approximates the distance of one environment step. As the agent gathers new experiences, we re-compute $\epsilon$. This is necessary because the encoder $E_\phi$, decoder $D_\theta$, and dynamics model parameters $M_\psi$ are constantly updated with the latest experience to ensure that ADM is not overfitted to only the demonstration data. In addition, we find the trajectory index $i$ and time step $t$ of the nearest demonstration using the ADM $d$:

$$i^*, t^* = \mathop{\arg\min}_{i,t} d(o', o_t^i), \qquad o_t^i \in \mathcal{D}. \tag{12}$$

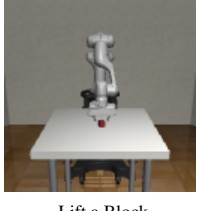 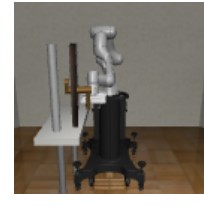 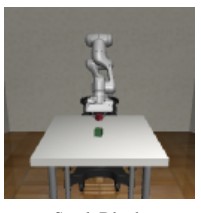 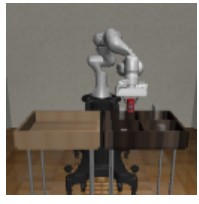

Lift a Block       Open a Door       Stack Blocks       Move a Can
5 demos        10 demos        10 demos        20 demos

Figure 4: LaNE achieves state-of-the-art sample efficiency in four Robosuite visual manipulation tasks. The RL agent observes RGB images from 2 cameras, one in the front (shown above) and the other on the gripper.

After we compute $\epsilon$, $i^*$, and $t^*$, we assign a dense reward $r_{\text{dense}}$ by augmenting the environment reward with a *task-progress-informed* exploration reward:

$$r_{\text{dense}} = \begin{cases} r + \alpha^{T_{i^*} - t^*} r_{\text{e}} & [d(o', o_{t^*}^{i^*}) \leq \epsilon] \wedge [o' \notin \mathcal{G}] \\ r & \text{otherwise.} \end{cases} \tag{13}$$

Here, the exploration reward $r_{\text{e}}^* = \alpha^{T_{i^*} - t^*} r_{\text{e}}$ is modulated by the expected step to success, which is the difference between the demo trajectory length $T_{i^*}$ and $t^*$. In the context of LfD, $r_{\text{e}}^*$ can be interpreted as a point estimate of the potential function at $o'$, echoing prior work in this domain [17, 20]. The discounting factor $\alpha$ is a hyper-parameter chosen independently from the RL discounting factor $\gamma$, and the nominal exploration reward $r_{\text{e}}$ is a constant.

Inspecting the augmented reward $r_{\text{dense}}$, we see that when $o'$ finds its nearest neighbor close to one of the successful terminal states, $o'$ is awarded almost the full nominal exploration reward $r_{\text{e}}$. When $o'$ is close to one of the earlier steps in a demonstration, $r_{\text{e}}$ is heavily discounted. Finally, if we are very far from any demonstration observation (relative to the distance threshold $\epsilon$), or if we are at the goal, the RL agent gets only the environment reward $r$ (case 2 in Eq. 13). LaNE is versatile because we can train a control policy to maximize $r_{\text{dense}}$ using any off-the-shelf RL algorithm.

### 4.4 Prioritized Replay and Value Clipping

We improve training efficiency and stability by performing prioritized replay and $Q$-value clipping. Prioritized replay is a standard tool when learning from demonstrations [19]: in each batch of $b$ transitions, we sample at least $p_d$ fraction from the demonstrations, where $b$ and $p_d$ are hyper-parameters. Conservative $q$-value estimates are also widely used to stabilize training [1, 43]. LaNE stands out because $r_{\text{dense}}$ derives upper and lower bounds on the $q$-value landscape, allowing us to use a clipped value target when performing temporal difference updates.

The definition of $r_{\text{dense}}$ in Eq. 13 contains the nominal exploration reward $r_{\text{e}}$, a constant hyper-parameter. We pick $|r_{\text{e}}| \leq |r_{\text{live}}|$ to obtain bounded $Q$-values. Under $r_{\text{dense}}$, a transition either receives a positive reward $r_{\text{done}}$ and terminates the episode or receives a non-positive reward $r_{\text{live}} + \mathbb{1} \cdot r_{\text{e}}^*$. The highest $Q$-value is achieved at task completion. On the other hand, in the worst case where the agent receives $r_{\text{live}}$ all the time and never succeeds, the $Q$-value is bounded below by $\sum_{t=0}^{\infty} \gamma^t r_{\text{live}} = \frac{1}{1-\gamma} r_{\text{live}}$. Thus, we obtain the following bound:

$$\frac{1}{1-\gamma} r_{\text{live}} \leq Q(o, a) \leq r_{\text{done}}.$$

## 5 Experiments

### 5.1 Simulated Manipulation Tasks

We solve four robot manipulation tasks from the Robosuite simulator [11]: lifting a block, opening a door, stacking blocks, and moving a soda can. The RL agent observes $128 \times 128$ RGB images from two cameras, one mounted on the robot gripper and one in front of the robot. The robot uses Operational Space Control – the agent predicts actions to change the robot's hand displacement,

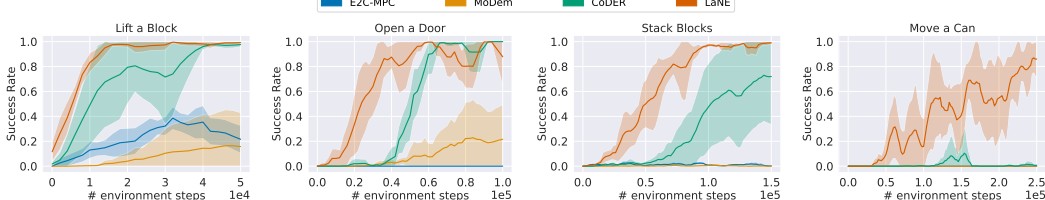

Figure 5: We compare LaNE with optimal control and state-of-the-art RL methods: E2C [41], MoDem [35] and CoDER [34]. Our method (red) consistently learns faster and converges to higher success rates than all three baseline methods.

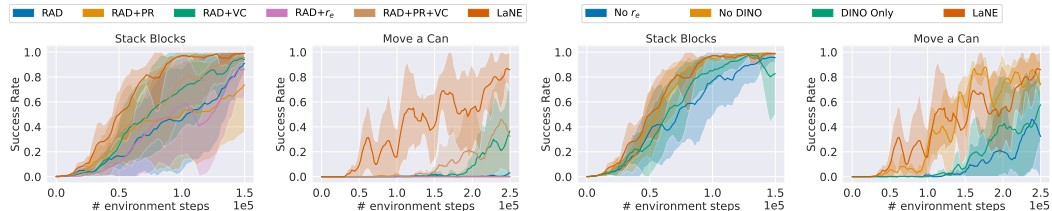

Figure 6: **Left:** Starting from RAD [8], we add PR (prioritized replay), VC (value clipping), and exploration reward $r_e$ to understand their contributions. PR and VC are necessary for stable learning but are inefficient. $r_e$ by itself results in unstable training. Hence, all components of LaNE are necessary. **Right:** We experiment with LaNE variants without pre-training (No DINO) or fine-tuning (DINO Only). While either variant improves from the baseline where no exploration reward is given, combining both results in the quickest exploration.

rotation, and gripper width for 7 degrees of freedom. Figure 4 shows one of the camera angles. The demonstrations come from a hand-coded controller that utilizes state information. The state is unavailable for the RL agent, who must learn from images alone.

We compare LaNE with optimal control, model-free, and model-based RL. For E2C [41], we run an MPC controller in the latent space to minimize a quadratic state cost towards the demonstrated goal state. CoDER [34], and MoDem [35] are state-of-the-art vision-based RL algorithms in the LfD setting. We initiate all methods with the same demonstrations and measure the evaluation success rates during training. Figure 5 shows the mean and standard deviation across five random seeds as a function of training environment steps. Our method outperforms all three baselines across all tasks while showing major advantages in the two more challenging tasks.

## 5.2 Ablation Studies

Our approach utilizes multiple techniques to maximize the utility of demonstrations and to speed up learning. We perform ablation studies to answer two key questions: **1.** Is reward augmentation really necessary, or are regularization tricks like prioritized replay (PR) and value clipping (VC) by themselves sufficient? **2.** LaNE learns an embedding space by fine-tuning a pre-trained computer vision model. Are both the fine-tuning step and the pre-trained DINOv2 model necessary?

For question 1, we start from a standard image-based RL algorithm RAD [8] (initialized with demonstrations) and add our key components, namely prioritized replay (PR), value clipping (VC), and the exploration reward $r_{\text{dense}}$. As shown by the left half of Figure 6, importance sampling and value clipping help stabilize training but are not efficient enough. $r_{\text{dense}}$ significantly contributes on top and allows the robot to complete the long-horizon task reliably much earlier in training.

For question 2, we experiment with two variants of LaNE, one without DINOv2 and one without fine-tuning. When LaNE trains without DINOv2, it initializes the encoder and decoder from scratch and learns to predict the images directly. When LaNE uses the pre-trained model and skips the fine-tuning steps, we directly use the Euclidean distance between their respective DINOv2 embedding vectors. The results in the right side of Figure 6 demonstrate that performing reward augmentation without pre-trained DINOv2 or fine-tuning helps, but combining both allows the most efficient learning. Notably, DINOv2 improves training convergence, as showcased in Figure 9.

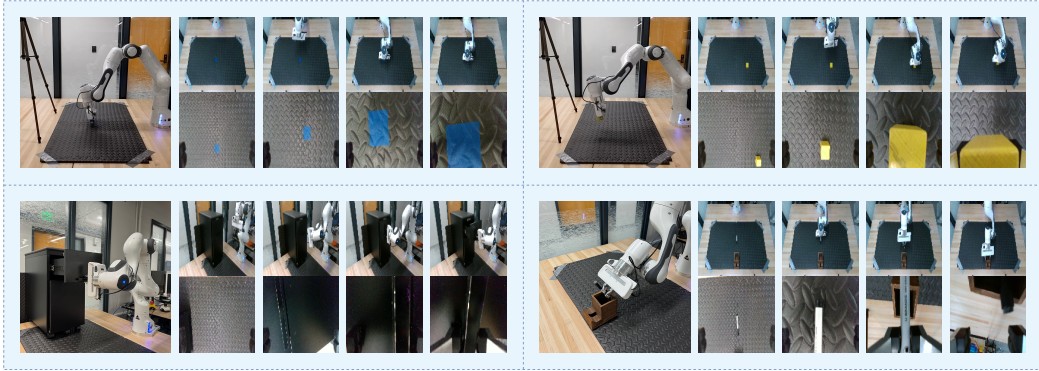

Figure 7: We deploy LaNE on four tasks with the Franka Panda robot: reach a fixed location, lift a block, open a drawer, and insert a pen. The RL agent observes two RGB images, one in front of the robot and one on its wrist. Trained from scratch, the robot achieves a 10/10 evaluation success rate with only a few demonstrations and less than one hour of learning.

## 5.3 Real Robot Experiments

As Figure 7 illustrates, LaNE enables efficient RL in four diverse tasks with the Franka Panda robot. From the easiest to the hardest, the tasks are Reaching a Fixed Goal, Opening a Drawer, Lifting a Block (random location), and Inserting a Pen. Demonstrations are collected via teleoperation from a mobile app, which uses inside-out tracking to stream the device's pose.

Results show that our method is extremely data-efficient, leading to task success with under an hour of training. Table 1 shows the number of demonstrations provided and training performance. For the most straightforward goal-reaching task, LaNE uses only one demo and completes the task for the first time in only 13 minutes. The agent finds a consistently successful policy quickly, only 8 minutes later. With five demonstrations, LaNE can learn more challenging tasks with high stochasticity or requiring precision, including lifting a block and inserting a pen. Overall, the results prove that our approach is outstanding in simulation and practical in the real world.

|  | Task | Reach | Drawer | Lift | Insert |
|---|---|---|---|---|---|
| Demos | Episodes | 1 | 1 | 5 | 5 |
|  | Steps | 12 | 18 | 103 | 135 |
|  | Time | 0:20 | 0:25 | 2:00 | 3:00 |
| CoDER First Success | Steps | N/A | N/A | N/A | N/A |
|  | Time | > 1h | > 1h | > 1h | > 1h |
| LaNE (Ours) First Success | Steps | 476 | 1054 | 1109 | 1854 |
|  | Time | 13:10 | 31:51 | 30:07 | 38:01 |
| LaNE (Ours) Convergence | Steps | 806 | 1929 | 1820 | 2927 |
|  | Time | 20:55 | 51:40 | 44:16 | 57:58 |

Table 1: We deploy LaNE on a Franka Panda arm to learn manipulation tasks requiring up to 7 degrees of freedom. Human demonstrations are provided via teleoperation, taking only a few minutes. LaNE trains every task to a 10/10 success rate from scratch under one hour of wall clock time. In comparison, our strongest baseline CoDER [34] fails to succeed even once during the first hour of training.

## 6 Conclusion

This paper presents LaNE, a data-efficient RL algorithm to learn sparse reward tasks from image observations by utilizing a few demonstrations. LaNE outperforms state-of-the-art benchmarks and enables under-an-hour RL training with a real robot. Our key innovation is to learn a latent dynamics model, which provides a temporally consistent embedding space. When a transition lands sufficiently close to a demonstration, we assign an extra task-progress-informed reward modulated by the distance to the goal. As such, we convert a sparse reward task to a task with dense proxy rewards, dramatically improving learning efficiency. Our work lends itself to exciting future directions. For example, we can leverage recent advancements in causal RL and counterfactual analysis [44, 45, 46] to determine the state in an expert demonstration that directly caused task success. This might further improve our search for the nearest demonstration and overall learning.

## Acknowledgements

This work was partly supported by ARL W911NF-21-1-0009, ONR 202102702AWD, DARPA ANSR: RTX CW2231110, and Lockheed Martin Corporation.

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

# 7 Appendix

## 7.1 Motivating Example

We motivate the need to learn a dynamics-aware embedding space: finding a good distance measure between states from their respective high-dimensional image observations is non-trivial. We use the PointMaze environment from the D4RL benchmark [47] to provide a clear illustration. In this environment, the controllable point mass is marked in green. As shown in Figure 8, we place the point at three positions in the maze such that state (a) is much closer to state (b) than state (c).

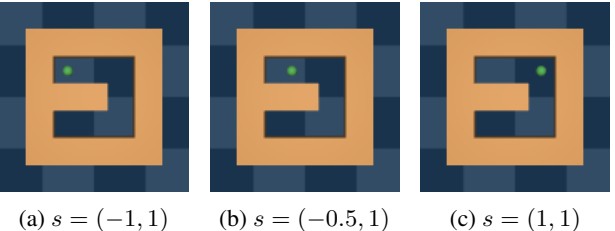

(a) $s = (-1, 1)$     (b) $s = (-0.5, 1)$     (c) $s = (1, 1)$

Figure 8: Observations from the PointMaze environment. The point mass in green is the controllable agent, whose location is indicated by $s$. As indicated by the agent's location, state $b$ is closer to state $a$ than $c$.

| Distance Measure $d$ | $d(a, b)$ | $d(a, c)$ | $d(a, c)/d(a, b)$ |
|---|---|---|---|
| Pixel $L^2$ | 2.835 | 3.303 | 1.165 |
| DINOv2 $L^2$ | 0.004 | 0.003 | 0.75 |
| Ground Truth $L^2$ | 0.5 | 1.5 | **3** |
| ADM (Ours) | 0.046 | 0.135 | **2.935** |

Table 2: Comparison across different distance measures in PointMaze. Distance in the pixel space and DINOv2 embedding space cannot capture the true relations of the underlying states. Whereas our proposed ADM can estimate the relative distance the most accurately.

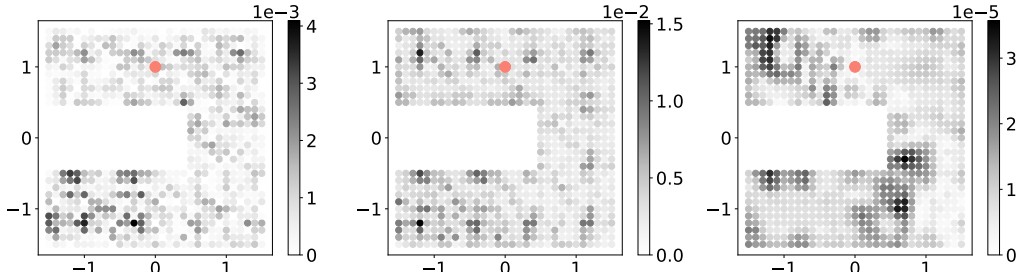

Figure 9: **Left:** ADM (Ours). **Middle:** DINOv2. **Right:** ADM without DINO. We plot the L2 distance in ADM and DINOv2 embedding space from each location in the maze to the pink location. In the ADM embedding space, computed distances match the environment dynamics, showing the lower left corner as the farthest due to the turn. In contrast, DINOv2 cannot identify the differences between states. Finally, if ADM is trained without DINOv2, it suffers from poor convergence due to difficulty in feature learning.

Table 2 compares two baseline distance measures with our proposed method ADM, trained from 10000 random interactions in the environment. For Pixel $L^2$, we directly compute the Euclidean distance. As expected, $d(a, b)$ and $d(a, c)$ are not distinguishable from the pixel-wise distance. Similarly, the pre-trained DINOv2 embeddings do not capture the transition information and fail to estimate the distance between states. ADM is the only method to capture the relative proportions of the ground-truth distances.

We visualize the quality of the learned embeddings in Figure 9, where we plot the estimated distance from all locations within the maze to a fixed location. We show that the embeddings learned by LaNE respect the dynamics of the maze, understanding that the point must go around the corner instead of

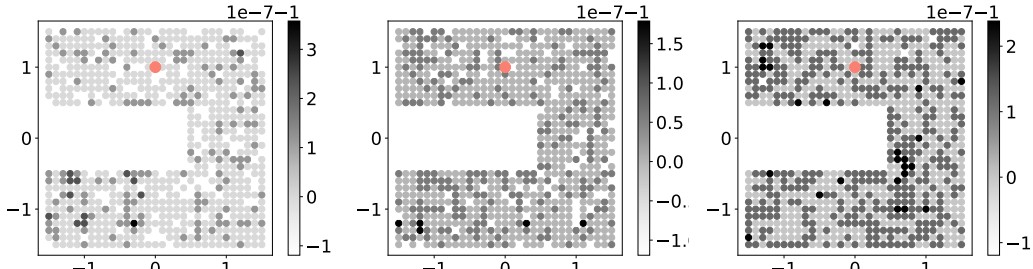

Figure 10: **Left:** ADM (Ours). **Middle:** DINOv2. **Right:** ADM without DINO. Instead of $L^2$ distance, we plot $d(v_1, v_2) := -\cos(v_1, v_2)$, the negative cosine distance between the embedding vectors. The scale $10^{-7}$ indicates that these vectors all point in similar directions. Hence, cosine similarity is not as informative as $L^2$.

through the wall. This simple experiment illustrates the benefits of using our latent space distance measure to quantify the task-relevant similarity between image observations.

Additionally, we are curious if cosine similarity, commonly used for vector retrieval, could be used in place of $L^2$ distance. As illustrated in Figure 10, all embedding vectors point in similar directions, making the cosine similarity less informative.

## 7.2 Visualization of Embedded Robot Trajectories

This section aims to provide a better intuitive understanding of the learned ADM embedding space. We visualize the learned embeddings in the real robot *reach* task by plotting their 2-D projections in Figure 11. Specifically, we plot a demonstration, a random successful, and a random unsuccessful episode. We observe a clear separation between successful and failed episodes: the successful episode closely follows the demonstration. In addition, we observe that the latent states move linearly as the agent progresses in the task.

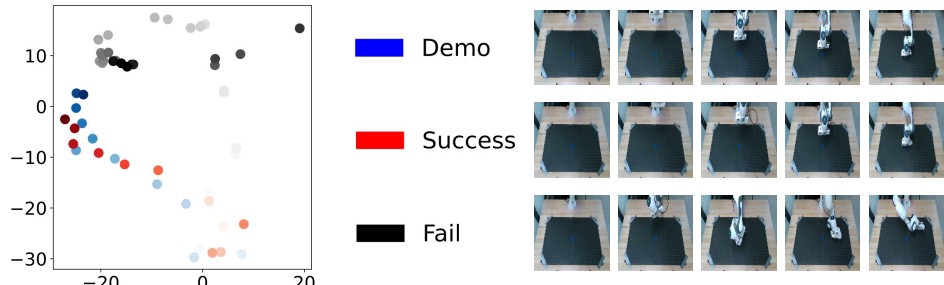

Figure 11: 2-D t-SNE of a demonstration, a successful and an unsuccessful trajectory on the Reach task. Colors change from light to dark as the episodes progress. The successful episode is mapped closer to the demonstration than the failed episode.

## 7.3 Comparison with Imitation Learning Methods

In this section, we compare LaNE with two powerful imitation learning algorithms, namely FISH [25] and PWIL [16]. FISH builds on a non-parametric base policy and trains an RL residual policy to minimize the earth mover's distance between the policy rollout and demonstration distribution. Similarly, PWIL uses RL to minimize an upper bound of the Wasserstein distance between rollouts and demonstrations. FISH uses an encoder pre-trained with behavior cloning as a feature extractor, whereas PWIL uses a TCC [37] model pretrained on demonstration trajectories.

As shown in Figure 12, both IL baselines fail to solve the Robosuite tasks given the same number of demonstrations as LaNE. One potential cause is that their feature extractors are learned solely from

demonstrations and hence struggle to generalize to OOD observations, which often occurs when the 7-DoF end-effector is allowed to rotate. On the other hand, these IL methods don't utilize any environment signals during RL training. Future works can explore the combination of sparse task rewards, online representation learning, and IL methods.

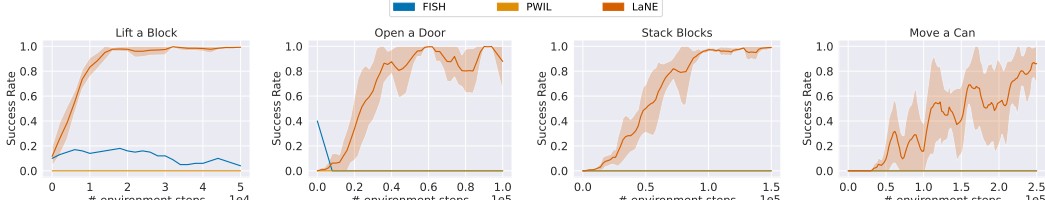

Figure 12: We compare LaNE with two imitation learning methods. Since the IL methods don't utilize the sparse environment rewards, they fail to learn the tasks from the limited number of demonstrations.

### 7.4 Other Ablation Studies

To justify the design choice of LaNE's exploration reward shown in Equation 13, we conduct ablation studies on variants of the exploration reward. 1. We set $\alpha = 1$ to verify if the discounting is necessary. 2. Alternatively, we use a simpler reward based on the L2 distance between the observation and its nearest-neighbor demonstration: $r_e = \max(1, \frac{\epsilon}{d(o', o_t^{i*})})$. This simplified reward is the inverse L2, scaled to the average distance between demonstration states, and clipped to make the $q$-value bound hold. Results on the left of Figure 13 show that our method outperforms both alternatives and holds a clear advantage in the long-horizon can-moving task.

Aside from our Augmentation-Invariant Distance Measure, we experiment with two other ways to learn image embeddings. 1. We verify the necessity of the local linearity constraint by swapping out the locally linear dynamics model with a fully connected neural network. 2. We perform contrastive learning with MoCo-v3 [48] on individual images without learning the dynamics. The right two plots in Figure 13 demonstrate that our method is advantageous over the alternative representation learning approaches.

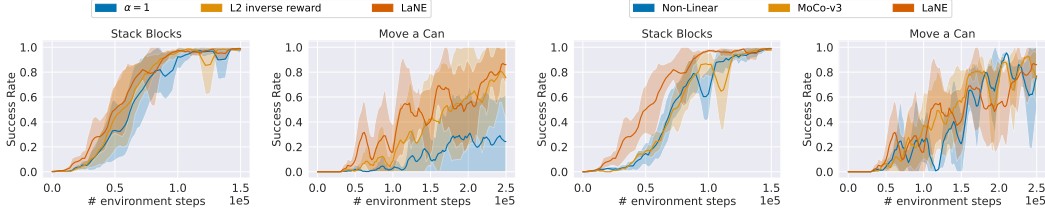

Figure 13: **Left:** We compare two other choices of exploration reward $r_e$. $\alpha = 1$ indicates the exploration bonus is not discounted based on the estimated distance to success. L2 inverse reward is a simple reward based on a state's estimated distance to its closest demo. Both alternatives are inferior to LaNE. **Right:** We compare two other representation learning methods to construct our distance measure. Non-Linear: We don't enforce local linearity in the forward model and use a fully connected network. MoCo-v3: We don't train a forward model but use contrastive learning on individual images. LaNE holds a clear advantage in the block stacking task. In move-a-can, LaNE is the earliest to achieve success.

### 7.5 Comparison with DreamerV2

We compare LaNE with DreamerV2, a state-of-the-art model-based RL method. Because DreamerV2 does not explicitly take demonstrations, we pre-fill its replay buffer with demonstration trajectories. In addition, we allow DreamerV2 to run for longer to get a better sense of its performance. As shown in Figure 14 below, it takes over five times as many environment steps for DreamerV2 to learn the Robosuite block-lifting task compared to LaNE. We note that DreamerV2 does not distinguish the pre-filled demonstrations from the regular episodes and thus fails to sample the sparse

reward often enough. Nevertheless, LaNE can also integrate with model-based RL methods to boost their sample efficiency in sparse-reward settings.

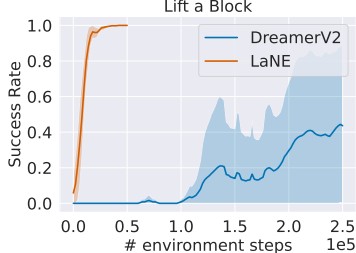

Figure 14: DreamerV2 takes over 5 times samples to learn the Robosuite Lift a Block task compared to LaNE because it does not distinguish between demonstration and regular episodes.

## 7.6 Robustness of LaNE Policies

We evaluate the out-of-distribution robustness of trained LaNE policies. Specifically, we push the block and the pen with a stick during policy execution, as illustrated below in Figure 15. We find that the policies are robust to small perturbations. For example, the agent can lift the yellow block when pushed, even when the stick and the operator's arm enter the camera view. Additionally, when the pen is moved to locations unseen during training, the robot can still pick it up.

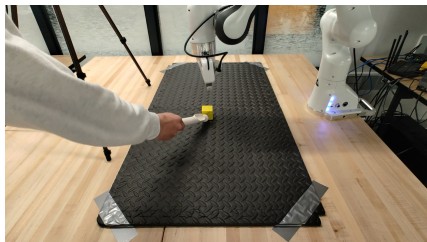 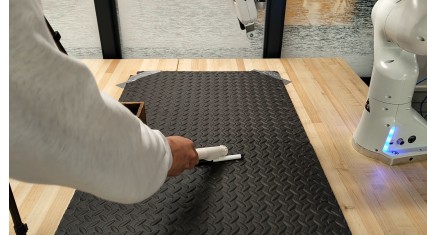

Figure 15: The learned LaNE policies are robust to small perturbations during evaluation, including moving objects, unseen initialization locations, and foreign objects.

## 7.7 Environment Details

*Maximum Episode Length:* For our simulated experiments in Robosuite, we set maximum episode lengths based on the difficulty of each task, as shown in Table 3. These numbers are slightly over the average steps the demonstrator takes to complete each task, leaving the RL agent plenty of time to finish. For our real robot experiments, all four tasks share the same maximum episode length of 30 steps. The episodes are terminated if the maximum length is reached or when the task is completed.

| Task | Lift | Door | Stack | Move a Can |
|---|---|---|---|---|
| # Max Episode Length | 40 | 80 | 80 | 120 |

Table 3: We choose the maximum episode length of each task slightly over the average steps the demonstrator took, giving the RL agent ample time to finish.

## 7.8 Real Robot Setup

### 7.8.1 Highly Compliant Real-time Controller

Because reinforcement learning requires a trial-and-error process [49], we expect the robot to make frequent physical contact with the environment. To eliminate safety hazards, the controller must be compliant with external forces. On the other hand, we want the robot to move swiftly so that each

RL step takes less time to execute. We develop a highly compliant real-time controller extended from the Cartesian Impedance Controller [50, 51].

At a high level, the end-effector tracks an equilibrium pose following a mass-spring-damper model. As the current end-effector pose deviates further from equilibrium, the robot asserts higher torque in the opposite direction. In the Cartesian space, we limit the maximum force exerted on the end-effector by the robot, preventing it from causing damage. In the joint space, we apply a counter torque when a joint gets close to its hardware limit. Combining these control rules builds a safety net around the robot for smoother RL training.

### 7.8.2 System Architecture

The robot uses two Intel Realsense cameras, one mounted on the end-effector and another in front of the robot. Our method only utilizes color images. Both cameras are connected via USB to an Nvidia GPU desktop, which runs inference and training for the RL agent. Specifically, the GPU desktop runs an OpenAI gym interface [10], with which the RL agent interacts. At each timestep, the agent chooses its action based on its policy: $a = \pi(o)$. The action $a$ consists of a displacement of the end effector position, change in roll/pitch/yaw, and open/close of the gripper. Next, the action selected by the RL agent is sent via ethernet to an Intel NUC, which directly interfaces with the Panda Robot. Specifically, the Intel NUC runs the Robot Operating System (ROS), where our real-time controller communicates with the hardware.

### 7.8.3 Demonstration Collection via Tele-Operation

A key aspect of our approach is to learn from a small set of human demonstrations efficiently. We build an application where a user can tele-operate the robot by moving and rotating a smartphone. Our iPhone application utilizes primitives from Apple's ARKit [52] to stream the position and orientation of the device to the PC controlling the robot.

During demonstration collection, a Python script translates the tracking data into gym actions and executes them on the real robot. The gym environment updates the robot's equilibrium pose to follow the phone's movement. These demonstration trajectories are stored on the Nvidia GPU desktop and used during training.

### 7.9 Hyper-parameters and Training Scheduling

We use a discounting factor $\gamma = 0.99$ and an exploration reward discount $\alpha = 0.98$ for all our experiments. The actor and critic learning rates are $10^{-3}$. The latent dynamics model learning rate is $4 \cdot 10^{-3}$. We use a training batch size of 128. We choose different fractions of demonstrations $p_d$ when we perform prioritized replay, as shown in the table below.

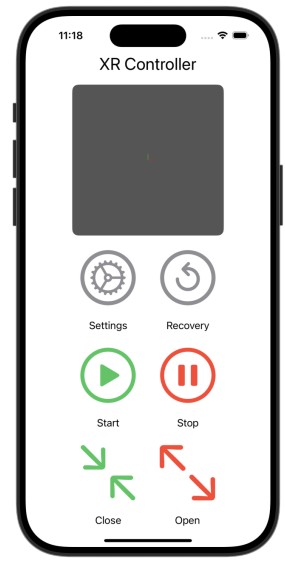

Figure 16: We develop a mobile app to stream the device pose to the PC. This allows us to collect demonstrations on the real robot arm quickly.

| Task | Lift | Door | Stack | Move a Can | Real Robot |
|------|------|------|-------|------------|------------|
| $p_d$ | 0.15 | 0.15 | 0.15 | 0.2 | 0.2 |

Table 4: We vary the fraction $p_d$ of demonstrations within each batch depending on the task we are training.

For simulated environments, we perform one Actor-Critic update for each environment step. After loading the demonstrations, we update the latent dynamics model every 5000 environment steps until convergence. For real robot experiments, we update the latent dynamics model every 30 steps the robot takes. Since each step takes about 0.5 seconds to execute on the real robot, we keep performing SAC updates in the background while the robot is in motion.

## 7.10 Neural Network Architectures

Our method consists of 6 components parameterized by neural networks, namely: model encoder $E_\phi$, model decoder $D_\theta$, locally-linear dynamics model $M_\psi$, RL encoder $E_{RL}$, Actor $\pi_{RL}$ and Critic $Q_{RL}$. The numbers below correspond to our specific setting where the latent space has 16 dimensions, and the action space has 7 dimensions. Our locally linear dynamics model predicts a low-rank approximation of $16 \times 16$ matrix $A$ using two 16-dimensional vectors $u$ and $v$, where $A = I + uv^T$.

```
Model Encoder (with DINOv2):
Input: 2 of 3x112x112 randomly cropped images
Pretrained DINOv2 variant: dinov2_vits14_reg
Concatenate 2 of DINOv2 embeddings
ReLU(Linear(out_features=512))
ReLU(Linear(out_features=512))
ReLU(Linear(out_features=512))
Linear(out_features=32)
Output: 16-dim mean + 16-dim log-std

Model Decoder (with DINOv2):
Input: 16-dim latent vectors
ReLU(Linear(out_features=512))
ReLU(Linear(out_features=512))
ReLU(Linear(out_features=512))
Linear(out_features=2 * 768)
Output: 2 of predicted DINOv2 embeddings

Model Encoder (No DINOv2):
Input: 6x112x112 randomly cropped images
ReLU(LayerNorm(Conv2D(6, 32, kernel=3, stride=2)))
ReLU(LayerNorm(Conv2D(32, 32, kernel=3)))
ReLU(LayerNorm(Conv2D(32, 32, kernel=3)))
Flatten()
ReLU(Linear(out_features=128))
ReLU(Linear(out_features=128))
Linear(out_features=32)
Output: 16-dim mean + 16-dim log-std

Model Decoder (No DINOv2):
Input: 16-dim latent vectors
ReLU(Linear(out_features=128))
ReLU(Linear(out_features=128))
ReLU(Linear(out_features=32768))
Reshape into 128x16x16
Upsample into 128x32x32
ReLU(Conv2D(128, 128, kernel=3, stride=1, pad=1)))
Upsample into 128x64x64
ReLU(Conv2D(128, 128, kernel=3, stride=1, pad=1)))
Upsample into 128x128x128
Conv2D(128, 6, kernel=3, stride=1, pad=1))
Output: 6x128x128 images

Locally-Linear Dynamics model:
Input: 16-dim latent vectors
ReLU(Linear(out_features=512))
ReLU(Linear(out_features=512))
ReLU(Linear(out_features=160))
Output: 16-dim vector u + 16-dim vector
        + 16x7 matrix B + 16-dim offset c

RL Encoder:
Input: 6x112x112 randomly cropped images
ReLU(LayerNorm(Conv2D(6, 32, kernel=3, stride=2)))
ReLU(LayerNorm(Conv2D(32, 32, kernel=3, stride=2)))
ReLU(LayerNorm(Conv2D(32, 32, kernel=3, stride=2)))
ReLU(LayerNorm(Conv2D(32, 32, kernel=3, stride=2)))
Flatten()
LayerNorm(Linear(out_features=32))
Output: 32-dim feature

Actor / Critic:
Input: 32-dim feature
ReLU(Linear(out_features=1024))
ReLU(Linear(out_features=1024))
Linear(out_features=14)
Output: Actor: 7-dim mean + 7-dim log-std; Critic: Q-value
```

