# OpenReview forum: "Accelerating Visual Sparse-Reward Learning with Latent Nearest-Demonstration-Guided Explorations"
_robot-learning.org/CoRL/2024/Conference — CoRL 2024_

### Official Review · Reviewer_qY4v · 2024-07-20
**Interesting idea, but some comparison and discussion missing**

**Originality:** 3
**Technical Quality:** 2
**Clarity Of Presentation:** 2
**Potential Impact:** 3
**Recommendation:** 2
**Confidence:** 3

**Review:**

Summary:

This paper presents Latent Nearest-demonstration-guided Exploration (LaNE), a method for solving sparse-reward robot manipulation tasks from image observations using a small number of demonstrations. The key ideas are:
- Learning a latent embedding space with locally linear dynamics to measure state proximity from images.
- Providing dense exploration rewards when the observation is considered similar to demonstrations in the latent space.

Strengths:

- The learning of a locally linear dynamic VAE and its use as a distance measurement is interesting. Similar methods have been used in deep model-based control methods. The way they use it for representation learning is innovative.
- Decent results on real robot tasks, demonstrating efficient learning of complex skills.

Weaknesses:

- The related work and comparison experiments still lack several potential methods (refer to Q1 and Q2 for more details). Especially given that the motivation is to create dense rewards for LfD by considering each step in the demonstration as a subgoal, similar ideas have been explored and should be compared.
- The generalizability of the learned latent space is concerning, especially when there are few demonstrations.
Potential brittleness to hyperparameter choices is not fully explored.

**Quality Of The Limitations Section:**

3

**Questions For Rebuttal:**

Q1: Comparison with Existing Dense Reward Methods

This work aims to accelerate Learning from Demonstration (LfD) with Reinforcement Learning (RL) by creating an augmented dense reward. However, designing dense rewards for LfD isn't always problematic. Previous works such as Temporal Cycle Consistency (TCC) and Time Contrastive Networks (TCN) have demonstrated the ability to learn stage representations of tasks and provide dense rewards, accelerating sparse reward LfD tasks. How does this work compare to or potentially complement these existing approaches? It would be valuable to see a comparative analysis or exploration of combining these methods to conclusively determine if the nearest retrieval augmented reward can significantly boost LfD in sparse reward settings.

Q2: Comparison with Existing Imitation Learning Reward Functions

The concept of designing a demonstration-guided reward is intriguing. However, there are existing imitation learning reward functions designed to accelerate LfD tasks. A notable example is Primal Wasserstein Imitation Learning (PWIL), which designs a dense reward by calculating greedy Wasserstein distance and has proven effective even when learning from a single demonstration. PWIL also evaluated using L2 distance as a reward to guide imitation. How does the proposed method compare to or differ from these existing approaches, particularly PWIL?

Q3: Role of Representation Learning Module

If the representation learning module (local linear dynamic VAE) is key to LaNE's effectiveness, it would be valuable to combine it with existing methods like PWIL. This combination could verify whether the local linear dynamic VAE plays a vital role in accelerating LfD across different reward design approaches. Have the authors considered or conducted such experiments?

Q4: Necessity of DINOv2

In Figure 6 (right), the ablation study seems to suggest that DINOv2 might not be necessary for LaNE. Could the authors elaborate on this finding? Additionally, for finding similar images, using cosine similarity instead of L2 distance might be more appropriate, as it's often used in deep image retrieval and may perform better for high-dimensional features. Have the authors considered this alternative metric?

Q5: Impact of Demonstration Quality and Multimodality

The augmented reward appears to constrain the policy to explore more around the demonstration region. This raises an interesting question: how do the quality and multimodality of demonstrations affect the policy learning performance in LaNE? Have the authors explored the robustness of their method to variations in demonstration quality or the presence of multiple valid ways to complete a task?

Q6: Discounting the exploration reward based on estimated distance to the goal

It is better to have ablation study to check how such design can accelerate/harm the learning.

**Robotics Focus:**

4

**Summary Of Paper:**

This work introduces a novel approach to designing dense rewards for Learning from Demonstration (LfD) tasks. The method, named LaNE, employs a locally linear dynamic VAE to encode features and generate a feature measurement metric, leveraging the vision foundation model DINOv2. During skill learning with Reinforcement Learning (RL), LaNE provides an auxiliary reward when the current image observation is deemed similar to the demonstrations. The authors assert that this augmented reward design significantly accelerates LfD tasks, offering a new perspective on bridging the gap between sparse demonstration data and efficient skill acquisition in robotics.

**Summary Of Recommendation:**

This work is clearly written, despite it shows the nearest retrieval augmentation can accelerate skill learning, it still lacks some related work comparisons and makes the conclusion less convincing..

---

### Official Review · Reviewer_9FLB · 2024-07-20
**Interesting work marred by insufficient comparison with baseline**

**Originality:** 3
**Technical Quality:** 3
**Clarity Of Presentation:** 3
**Potential Impact:** 3
**Recommendation:** 3
**Confidence:** 4

**Review:**

## Summary of the work

In this work, the authors present a new way of doing demo-guided RL by first trying to learn a latent space encoder that obeys linear transition dynamics, and then using this latent encoder to find nearest neighbors to give denser rewards to accelerate the RL exploration phase.

In the first phase, the authors fine-tune a foundational encoder model (DINO) with a latent dynamics based encoder objective function. It tries to enforce that the dynamics in the latent space follows a linear model. With this objective, the trained encoder is able to provide a distance metric in the representation space, which is useful for the next step.

In the second phase, with the trained encoder, the authors design a novel dense reward function for demo-guided RL. To summarize, this dense reward checks if the achieved state is "close" to a goal/reward generating state in the latent distance metric (and also temporally), and if so, adds an appropriately scaled dense reward to augment the sparse reward.

The authors show experiments on 4 simulated setups and 4 real world, real robot tasks. They compare against RL baselines (*more on this later*) and show that their method outperform their baselines.

## Strengths
The author's proposed latent dynamic objective seems to be an interesting one, and I would be interested in seeing further analysis of it in non-RL setups.

## Weaknesses
1. Despite the algorithm being somewhat convoluted, I think the biggest reason why I am suggesting a refusal is because **the baselines are totally irrelevant!** This work in my mind is categorized as demo-guided RL, and should be compared against an entirely different class of algorithms such as DAC [1], ROT [2], or FISH [3]. In particular, [2-3] actually make a stronger claim than this paper, of doing real world RL with a much smaller amount of demo while solving tasks in an hour. Why are the authors using methods that are demo-free?
2. There is an arbitrary amount of demo data used for each task with no justification.
3. The ablations are also not complete: why not analyze the impact of the extra dynamic-guided fine-tuning? That's a big part of the paper. What happens if we use simple image-space encoder fine-tuning like MoCo-v3?

[1] Kostrikov, Ilya, et al. "Discriminator-actor-critic: Addressing sample inefficiency and reward bias in adversarial imitation learning." arXiv preprint arXiv:1809.02925 (2018).
[2] Haldar, Siddhant, et al. "Teach a robot to fish: Versatile imitation from one minute of demonstrations." arXiv preprint arXiv:2303.01497 (2023).
[3] Haldar, Siddhant, et al. "Watch and match: Supercharging imitation with regularized optimal transport." Conference on Robot Learning. PMLR, 2023.

**Quality Of The Limitations Section:**

2

**Questions For Rebuttal:**

1. Why compare with non-demo guided RL methods if there are so many good demo-guided RL methods out there? This seems like an obvious oversight, what am I missing?
2. What happens if we use simple image-space encoder fine-tuning like MoCo-v3?
3. Can this pretraining improve, say, imitation learning methods?

**Robotics Focus:**

4

**Summary Of Paper:**

Train a dynamic aware autoencoder, and using nearest neighbor in its latent space, give extra reward for exploration

**Summary Of Recommendation:**

Given the rebuttal, I updated my assessment to a weak accept

---

### Official Review · Reviewer_ic1h · 2024-07-21
**The paper presents a novel visual RL method, LaNE, with promising results in robotic manipulation tasks, but requires additional ablation studies, clarification of methods, and open-sourcing of code for thorough evaluation.**

**Originality:** 3
**Technical Quality:** 3
**Clarity Of Presentation:** 3
**Potential Impact:** 2
**Recommendation:** 3
**Confidence:** 5

**Review:**

Strengths:
* Innovative Approach: Many RL studies typically rely on domain-specific and human-crafted reward functions. This paper introduces a sample-efficient RL approach that enables learning from sparse rewards by additionally rewarding an agent for exploring near the demonstrations.
* Robust Performance: The authors demonstrate the robustness of their method through performance evaluations in both simulated and real robot environments.
* Real-World Application: The proposed method, LaNE, is shown to train a real Franka Panda robot from scratch within an hour using only a few demonstrations, highlighting its practical applicability.

Weakness:
* Dependence on State Tracking: To obtain sparse rewards in a real robot setting, the complete or partial system state (e.g., positions and velocities of a gripper and objects) must be reliably tracked. To fully exploit the advantages of using image observations, it is necessary to present experimental results without sparse rewards and only with task-progress-informed rewards, which the authors additionally provide.
* Code Availability: The authors need to open-source their code to allow reviewers and other researchers to objectively validate the performance of their method.

**Quality Of The Limitations Section:**

2

**Questions For Rebuttal:**

Issue 1: The authors train a latent dynamics model in a linear system manner. Why not train a neural network for the dynamics model, which can handle non-linearity? Please provide an additional ablation study comparing the linear dynamics model to a non-linear dynamics model.

Issue 2: Using sparse rewards remains an issue in real robot environments. Please provide additional experiments using only task-progress-informed exploration rewards without relying on sparse rewards.

Issue 3: In Fig. 5, the performance of MoDem presented by the authors is significantly worse compared to the performance reported in the MoDem paper. To objectively evaluate the performance comparison, we request the authors to open source their code. The authors can also perform a comparative evaluation with MoDem-v2 in a real robot environment.

Issue 4: In Fig. 6, the authors present ablation studies on adding PR (prioritized replay), VC (value clipping), and exploration reward $r_e$, based on RAD. However, some combinations are missing, such as RAD+VC and RAD+$r_e$. Since the effect of $r_e$ is expected to be the most pronounced, we request additional ablation studies for each combination.

Issue 5: In Fig. 6, there is little performance difference between w/o DINO and w/ DINO, which may suggest that a CNN-based encoder and decoder are sufficient without using DINO. To confirm this, a visualization of the representation for $z$ is needed for both w/o DINO and w/ DINO conditions.

**Robotics Focus:**

4

**Summary Of Paper:**

In this paper, the authors present a visual RL method called LaNE, designed to efficiently learn a policy for robotic manipulation tasks with sparse rewards from image observations and a few demonstrations. LaNE achieves superior sample efficiency by learning an embedding space to quantify state proximity and reward explorations near the demonstrations. Experiments demonstrate that LaNE can train a real Franka Panda robot from scratch within an hour using only a few demonstrations.

**Summary Of Recommendation:**

The paper is well-written and presents a novel visual RL method, LaNE, that efficiently learns policies for robotic manipulation tasks with sparse rewards using image observations and limited demonstrations. However, several issues need to be addressed, including providing additional ablation studies comparing linear and non-linear dynamics models, experimenting with task-progress-informed exploration rewards without sparse rewards, and addressing discrepancies in the performance of MoDem. Additionally, open-sourcing the code and providing visualizations of the representation for $z$ with and without DINO would greatly enhance the reproducibility and robustness of the results. Addressing these issues would significantly strengthen the contribution and impact of the work.

---

### Author Rebuttal · Authors · 2024-08-13

We appreciate all reviewers for taking the time to understand the core of our work and provide insightful feedback! In addition to the rebuttal below, we will address each reviewer's comment individually.

---

## Clarification of Problem Setting

We adopt the same setting as MoDem and CoDER (aka FERM). In addition to the demonstrations, the agent utilizes (sparse) task rewards. This allows LaNE to solve tasks where imitation learning fails. We tried FISH and TCC+PWIL, which failed to solve the tasks. We invite the reviewer to check the detailed results in Section 7.3 and the attached baseline code.

The IL experiments are placed in the appendix because they compete at a disadvantage. Future works can explore combining these distribution-matching objectives with (sparse) external rewards and online representation learning. This is an exciting direction but way beyond this paper's scope.

---

## Note on Task Difficulty

It might appear counter-intuitive why the baseline results are poor. Our 7-DoF manipulation tasks pose significant challenges to policy and representation learning. When the end-effector rotates slightly, the gripper camera observation changes drastically and can quickly go outside the demonstration distribution. We invite the reviewers to check the task videos and notice the aggressive camera angle changes.

---

## Paper Revision and Supplementary Material Updates

All significant changes to the original manuscript are colored in blue. We value transparency in the review process. Hence, we also provide the source code and experiment logs for the MoDem, FISH, and TCC+PWIL.

---

### Decision · Program_Chairs · 2024-09-04

**Decision:**

Accept

**Comment:**

Strengths
- Interesting method
- Nice evaluation in simulation and on a real robot

Weaknesses
- Doubts about limited setting (sparse rewards)
- Missing details and justifications
- Doubts about results of some of the comparisons
- Baselines don't seem to be the most relevant ones

# After Discussion Phase
Both reviewers ic1h and 9FLB were very happy with the replies and additional material/experiments and accordingly updated their scores. Reviewer ic1h still believes that adding a feature visualization to confirm the effectiveness of using DINOv2 would make the paper stronger. Reviewer qY4v unfortunately did not react, but also those comments were well addressed and the requested experiments and ablations added (with results further supporting the method).